# Designing and validating a Markov model for hospital-based addiction consult service impact on 12-month drug and non-drug related mortality

Caroline A. King[1]*, Honora Englander[2], P. Todd Korthuis[2], Joshua A. Barocas[3], K. John McConnell[4], Cynthia D. Morris[5], Ryan Cook[2]

1 Dept. of Biomedical Engineering, School of Medicine, Oregon Health & Science University, Portland, OR, United States of America, 2 Department of Medicine, Section of Addiction Medicine, Oregon Health & Science University, Portland, OR, United States of America, 3 Section of Infectious Diseases, Boston University School of Medicine and Boston Medical Center, Boston, MA, United States of America, 4 Center for Health Systems Effectiveness, Oregon Health & Science University, Portland, OR, United States of America, 5 Department of Medical Informatics and Clinical Epidemiology, Oregon Health & Science University, Portland, OR, United States of America

* kingca@ohsu.edu

**Data Availability Statement:** Researchers can access data from this study, but need to be approved by OHSU's Institutional Review Board

## Abstract

### Introduction

Addiction consult services (ACS) engage hospitalized patients with opioid use disorder (OUD) in care and help meet their goals for substance use treatment. Little is known about how ACS affect mortality for patients with OUD. The objective of this study was to design and validate a model that estimates the impact of ACS care on 12-month mortality among hospitalized patients with OUD.

### Methods

We developed a Markov model of referral to an ACS, post-discharge engagement in SUD care, and 12-month drug-related and non-drug related mortality among hospitalized patients with OUD. We populated our model using Oregon Medicaid data and validated it using international modeling standards.

### Results

There were 6,654 patients with OUD hospitalized from April 2015 through December 2017. There were 114 (1.7%) drug-related deaths and 408 (6.1%) non-drug related deaths at 12 months. Bayesian logistic regression models estimated four percent (4%, 95% CI = 2%, 6%) of patients were referred to an ACS. Of those, 47% (95% CI = 37%, 57%) engaged in post-discharge OUD care, versus 20% not referred to an ACS (95% CI = 16%, 24%). The risk of drug-related death at 12 months among patients in post-discharge OUD care was 3% (95% CI = 0%, 7%) versus 6% not in care (95% CI = 2%, 10%). The risk of non-drug related death was 7% (95% CI = 1%, 13%) among patients in post-discharge OUD treatment,

(irb@ohsu.edu), and undergo training to access the data through the Center for Health Systems Effectiveness. Data cannot be shared publicly because information is potentially identifiable and legal data use agreements do not permit data sharing without appropriate training. Requests for access and training can be sent to Dr. John McConnell, Director for the Center for Health Systems Effectiveness, at mcconnj@ohsu.edu.

**Funding:** This research was supported through grants from the National Institutes of Health, National Institute on Drug Abuse (UG1DA015815, UG3DA044831). Grant UL1TR002369 provided support of REDCap, the web application this study used for data collection. Caroline King was supported by the National Center for Advancing Translational Sciences, National Institutes of Health, through Grant Award Number TL1TR002371 and the National Institute On Drug Abuse of the National Institutes of Health under Award Number F30DA052972. The content is solely the responsibility of the authors and does not necessarily represent the official views of the NIH. The funders had no role in study design, data collection and analysis, decision to publish, or preparation of the manuscript.

**Competing interests:** Dr. Korthuis serves as principal investigator for NIH-funded studies that accept donated study medication from Alkermes (extended-release naltrexone) and Indivior (buprenorphine). This does not alter our adherence to PLOS ONE policies on sharing data and materials.

versus 9% not in care (95% CI = 5%, 13%). We validated our model by evaluating its predictive, external, internal, face and cross validity.

## Discussion

Our novel Markov model reflects trajectories of care and survival for patients hospitalized with OUD. This model can be used to evaluate the impact of other clinical and policy changes to improve patient survival.

## Introduction

Drug overdose is the leading cause of unintentional injury death in the United States [1]. Among people with opioid use disorder (OUD), an estimated 20% eventually die of drug overdose [2], but cardiovascular diseases, cancer, and infectious diseases also contribute to mortality rates. Patients with OUD who are hospitalized for OUD-related and other diagnoses are often medically complex and face life-threatening illnesses. These patients experience higher mortality rates than hospitalized patients with similar conditions [2].

Hospitalization is a vulnerable time for patients with OUD. People with OUD may leave the hospital before completing recommended medical therapy if withdrawal symptoms are untreated [3]. People who withdraw from opioids have lower drug tolerance and increased risk of drug overdose after discharge in the absence of treatment for OUD [4–6]. Medications for opioid use disorder (MOUD), including methadone, buprenorphine and naltrexone, can reduce the risk of death from opioid overdose in patients with OUD [7]. These medications work as opioid receptor full agonists (methadone), partial agonists (buprenorphine), or antagonists (naltrexone) [8]. Despite the success of MOUD to reduce opioid overdose deaths, most hospitalized patients with OUD are not started on MOUD [9, 10], though, when offered, nearly three-quarters of patients with OUD choose to start MOUD [11]. Interventions to improve initiation of MOUD among hospitalized patients are urgently needed [12].

Addiction consult services (ACS) are an emerging intervention to engage hospitalized patients in care and meet patient-driven goals for substance use treatment [13]. Typically, they include care from an interprofessional team that may include medical providers, social workers, nurses, and alcohol and drug counselors [14]. Some intentionally include people with lived experience in recovery [15–17]. ACSs typically address the needs of people who use any substance (for example, stimulant, alcohol, and opioids). Care includes comprehensive assessments, withdrawal management, medication treatment, psychosocial and harm reduction interventions, and efforts to support patient engagement and linkage to care across settings. ACSs commonly also provide staff education and patient advocacy [14, 18, 19]. Evaluation of ACS demonstrates improved engagement in post-hospitalization treatment and decreased substance use [12, 13]. However, assessing the effect of ACS using gold-standard study designs is challenging because of the costs and logistical challenges associated with multi-site, cluster-randomized trials. Additionally, it can be difficult statistically to assess distal, rare outcomes like drug-related mortality in the context of a hospital-based intervention. We consequently do not know how ACSs affect post-discharge drug-related mortality or non-drug related mortality for patients with OUD.

Simulation modeling allows researchers to rapidly test different care delivery scenarios and capture robust estimates of study outcomes, which can support healthcare system decision-making and answer salient clinical questions in the midst of the opioid overdose epidemic.

Modeling inpatient care scenarios can guide healthcare systems in addressing a rapidly evolving epidemic more quickly and adaptively than randomized trials. Simulation modeling has previously been used to estimate prevented overdose deaths from the expansion of naloxone distribution [20–22], the progression of opioid addiction [23], and the implementation of safe-injection sites [24]. The objective of this study was to design and validate a Markov model that estimates the impact of ACS care on 12-month mortality among hospitalized patients with OUD.

## Methods

### Setting and study design

Oregon Health & Science University in Portland, Oregon is home to an inpatient ACS, the Improving Addiction Care Team (IMPACT). IMPACT is a hospital-based service that utilizes an interdisciplinary team of physicians, advanced practice providers, social workers, and peers with lived experience in recovery to support non-treatment seeking adults with substance use disorder. Patients are eligible to be referred if they have known or suspected substance use disorder (SUD), other than tobacco use disorder alone. IMPACT conducts substance use assessments, initiates medication-based treatment (including buprenorphine, methadone and extended release naltrexone for OUD) and behavioral treatment where appropriate, and connects patients to post-discharge SUD treatment. IMPACT utilizes a harm reduction approach and integrates principles of trauma-informed care. Previous research describes IMPACT's design and evaluation [11, 12, 16–18, 25, 26]. Notably, IMPACT is the only comprehensive ACS in Oregon, though a few hospitals offer MOUD initiation during hospitalization.

We developed and validated a Markov model to estimate the impact of ACS care on 12-month mortality among hospitalized patients with OUD (Fig 1). We organize our methods in the order of completion: first, we decided on model structure; next we used available data to populate the model; and finally, we validated the model. As such, we describe: 1) model structure, 2) model data, and 3) model validation.

The Oregon Health & Science University's Institutional Review Board approved this study and waived the requirement for informed consent (#00010846).

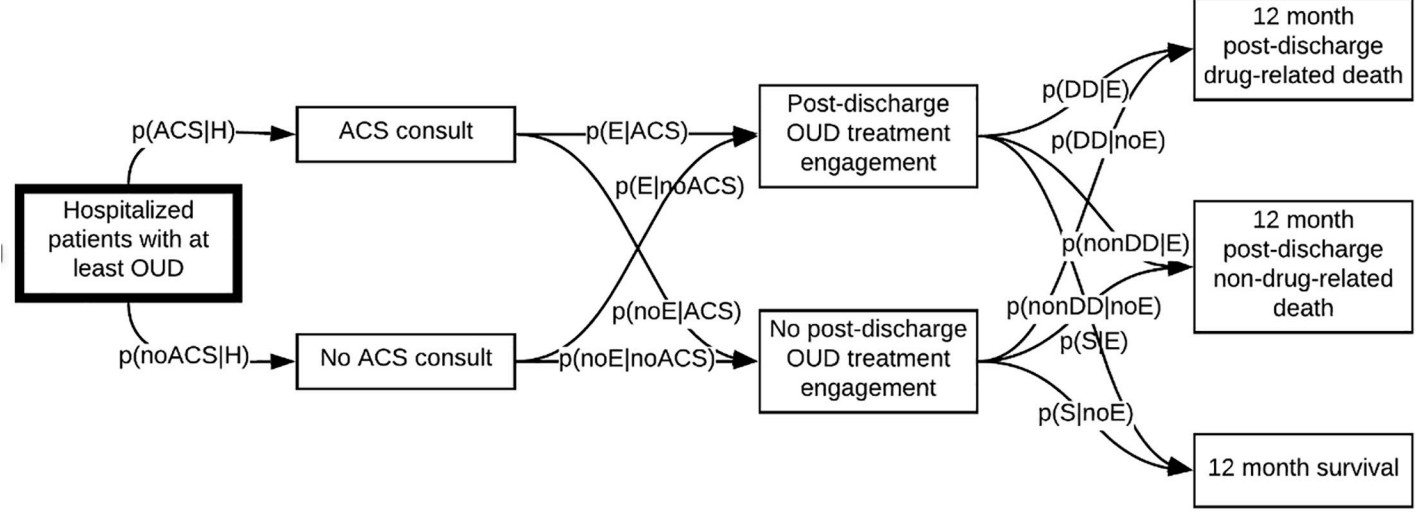

**Fig 1. Markov model of hospital-based addiction care in Oregon, 2015–2018.**

## 1) Model structure

Our model reflects key components of care as patients move through hospitalization, discharge, and post-hospital time periods. The model has the following components: ACS consult, post-discharge OUD treatment engagement, and 12-month post-discharge drug related death, non-drug related death, and survival.

**ACS referral.** Once patients are hospitalized, they can be referred to ACS care. ACSs exist across a growing number of North American hospitals.

**Post-discharge OUD treatment engagement.** We used a modified Healthcare Effectiveness Data and Information Set (HEDIS) measure of engagement to stratify for post-discharge OUD treatment engagement. The original measure requires that patients initiate treatment and have two or more additional alcohol or drug services or medication for OUD within 34 days of initiation [27]. Recent research has shown that evidence-based MOUD has superior outcomes in preventing mortality and decreasing opioid use [7]. For this reason, we defined post-discharge OUD treatment engagement as: 1) at least two filled prescriptions for buprenorphine, extended-release naltrexone, or methadone from an Opioid Treatment Program in the 30 days following hospital discharge, or 2) a prescription for extended-release naltrexone or buprenorphine that covered 28 of the 30 days post-hospital discharge [28].

**12-month mortality.** At twelve months, deaths are classified as drug related versus non-drug related (including as circulatory, neoplasm, infectious, digestive (including alcohol-related liver disease), external (including suicide and unintentional injury), respiratory, endocrine, and other) by ICD-10 mortality codes described by Hser et al. [2].

## 2) Model data

Our Markov model could be used in any setting with patients hospitalized with OUD where data exists for recalibration. We populated our model with data from Oregon Medicaid claims data and expert opinion, described below, to reflect care from an addiction consult service in Portland, Oregon, and its impact on post-discharge drug and non-drug related mortality.

We had multiple goals in using data to populate our model. We needed a dataset of patients, where some patients were referred to ACS and some were not. Then, we needed to be able to match ACS patients to controls as one way to account for some confounding. We needed the dataset to follow both patients referred to ACS, and those not, through 12 months after hospital discharge. Finally, we needed the dataset to have additional covariates to control for additional confounding, which we planned to do via logistic regression models at each transition point. Below, we describe the merging of OHSU's ACS dataset with Oregon Medicaid data and Vital Statistics data, to achieve our goal for a dataset for the model population. Because we wanted to incorporate national estimates into our dataset, we used Bayesian logistic regression to integrate expert opinion into our estimates. We describe each of these steps below.

## Participants

To generate probability of ACS referral and post-discharge treatment engagement, we used Oregon Medicaid claims data to identify patients with OUD hospitalized at least once from April 2015 through August 2018, including IMPACT patients. Because OHSU IMPACT was the only ACS in Oregon during the study window, we used IMPACT registry which tracks all referrals to identify patients with Oregon Medicaid who were referred to ACS. To generate probability of 12-month mortality, we utilized mortality data from Oregon Vital Statistics through December 31, 2018; thus, we included only patients admitted through January 1, 2018 to allow 12 months of follow-up time. [Patients were eligible for inclusion if they were over 18

years old and had an ICD-9 (304.*) or ICD-10 (F11*) diagnosis of OUD during a hospital admission.

**Cohorts for transition points.** We defined three cohorts for our analyses utilizing Oregon Medicaid data. First, we included all patients who met eligibility criteria in analysis for our first transition, referral to ACS. Then, we used a matched cohort of three controls to one IMPACT patient for our post-discharge OUD care engagement and mortality analyses. We matched without replacement on hospital admission quarter and admission number, including one admission per person.

## Transition data

For ACS referral, we identified all hospitalized patients with OUD in Oregon during the study period, and then identified the subset who were referred to the ACS. For post-discharge OUD treatment engagement, we used Oregon Medicaid claims data to identify if patients met the modified HEDIS engagement measure in the 30 days following hospital discharge. For 12-month mortality, we used Oregon Vital Statistics data to identify deaths in our cohort during the study period through December 31, 2018. For mortality models, the cohort was limited to include only participants seen before January 1, 2018 to allow for 12 months of follow-up time for all participants. We classified deaths as drug related versus non-drug related as indicated above. We manually reviewed deaths that were not captured by these codes and reclassified to fit into drug versus non-drug related categories.

## Transition probabilities

We used a Bayesian approach to obtain transition probabilities for our Markov model using Oregon data. In short, we integrated national expert information with estimates from Oregon data for each of our three transition steps: ACS referral, post-discharge MOUD, and 12-month mortality. We also adjusted for confounding at each transition point. Bayesian logistic regression allowed us to accomplish this goal. We ran logistic regression models for each transition point, using the transition as the outcome (i.e. an outcome of 12-month post-discharge mortality) and the prior step as the primary covariate of interest (i.e. 30-day post-discharge MOUD), adjusting for all other covariates in the model. We extract a marginal probability from this logistic regression model- this is our Bayesian likelihood. We used information from experts in addiction as our prior. The Bayesian approach allows the integration of the prior and likelihood to estimate a posterior probability, which we use as our posterior probability.

**Bayesian priors via expert elicitation.** Because of the novelty of ACS, few published papers existed from which we could have derived prior estimates of transition probabilities for Bayesian analysis. Thus, we used expert elicitation to capture prior information for our models. The Bayesian process helped account for some for some of the uncertainty that comes from incorporating expert opinion. We identified important covariates at each transition point, including age (in years), gender (female/male), race (White/not White/unknown), ethnicity (Hispanic/Not Hispanic), concurrent alcohol use disorder (yes/no), concurrent stimulant use disorder (yes/no), hospital length of stay (in days), rural residence (yes/no), filled at least one prescription for medication for OUD in the month before hospital admission (yes/no), previously admitted to the hospital (yes/no), and Chronic Illness and Disability Payment System (CDPS) Score (continuous). The engagement model also included referral to an ACS (yes/no). The mortality models included engagement in care after discharge (yes/no) and filled a naloxone prescription in the 30 days after hospital discharge (yes/no).

We used a clinical-vignette design to ask providers about the relevance of covariates on patient outcomes. To do this, participants provided a probability estimate for different events: referral to an ACS, post-discharge engagement, and mortality.

For example, a vignette could read:

*"The patient is a young White man with OUD and AUD. He was in the hospital for several days. He was on medication for OUD at admission. He had never previously been admitted to the hospital. He has many comorbidities. He is not from a rural area. What is the probability he engaged in post-discharge treatment for OUD within 30 days of discharge?"*

Experts evaluated 16 (referral to ACS), 17 (engagement) and 18 (mortality) vignettes selected from an optimal experimental design generated for each model [29]. From the optimal design, we chose a subset of the vignettes that were substantially different from one other for ease of interpretability and to maximize the information gathered about each covariate.

As part of our IRB-approved research, study authors (HE, PTK) generated lists of experts in addiction consult services and hospital-based addiction treatment in general in the United States. Each participant took only one survey. We aimed to recruit at least five participants for each survey, with a goal of at least three responses per survey. For the referral to ACS survey, we also asked participants to refer hospitalists at their institutions to complete the survey, as hospitalists are frequently providers who refer patients to ACS. Ultimately, six participants took the ACS survey (6 of 11, 54.5%), four took the engagement survey (4 of 5, 80%), and three took the mortality survey (3 of 8, 37.5%).

After surveying expert participants, we calculated the mean and identified the minimum and maximum ratings. We then numerically fit beta distributions to those quantities using differing "confidence levels" [30]. Then, we updated our priors with the information from data about our cohort described above. We estimated marginal probabilities over observed cases using fitted Bayesian logistic regression models at each transition point [31].

**Bayesian logistic regression models.** We used the transformed prior information from expert surveys and Oregon Medicaid cohort data to fit Bayesian logistic regression models at each transition point. Models were fit using Markov Chain Monte Carlo methods [32]. We sampled each parameter 10,000 times with 2000 burn-in chains. We used multiple metrics to assess model convergence. First, we used Gelman and Rubin's potential scale reduction factor; all values in all models equal 1.0. Values close to 1.0 are suggestive of convergence. Effective sample sizes all approximated the number of posterior draws requested. All model trace plots appear to have a caterpillar-like distribution, and there were no divergent transitions. Autocorrelation plots for all parameters suggest low autocorrelation. We used the package Shiny Stan to evaluate Bayesian model fit [33].

We tested different prior information strengths: first, using a cohort sample size method, where the prior information equivalates a percent of the study sample size (0.1%, 1%, 5% and 10%); second, using a confidence interval method, where we fit beta distributions to the range of survey responses, and then used the maximum and minimum values as borders for 80%, 85%, 90%, and 95% confidence intervals. We picked the best-fit model using Pareto smoothed importance-sampling leave-one-out cross validation using the loo package in R where lower expected log predictive density values indicate a better model fit [34]. We also prioritized models where Pareto k diagnostic values had at least good reliability for all estimates.

We used mcmcObsProb in the BayesPostEst package [35] to estimate marginal transition probabilities over observed cases with the fitted Bayesian logistic regression models. We created prior-posterior plots using ggplot2 [36].

**Table 1. Participant demographics.**

|  | All patients n = 8,450 | Seen by ACS n = 265 | Not Seen by ACS n = 8,185 | p-value |
|---|---|---|---|---|
| **Age** Years | 44.5 (15.4) | 39.5 (0.77) | 44.6 (0.17) | <0.001 |
| **Gender** Male | 3,632 (43.0%) | 159 (60.0%) | 3,473 (42.4%) | <0.001 |
| **Race** White | 5,919 (70.1%) | 169 (63.8%) | 5,750 (70.3%) | 0.034 |
| Not White | 543 (6.4%) | 16 (6.0%) | 527 (6.4%) | |
| Unknown race | 1,988 (23.5%) | 80 (30.2%) | 1,908 (23.3%) | |
| **Ethnicity** Hispanic | 299 (3.5%) | 10 (3.8%) | 289 (3.5%) | 0.002 |
| **Alcohol use disorder** | 306 (3.6%) | 14 (5.3%) | 322 (3.9%) | 0.269 |
| **Stimulant use disorder** | 689 (8.2%) | 41 (15.5%) | 642 (7.8%) | <0.001 |
| **Length of stay (days)** | 6.6 (11.2) | 14.9 (0.97) | 6.4 (0.12) | <0.001 |
| **Rural residence** | 2,234 (26.4%) | 32 (12.1%) | 2,202 (26.9%) | <0.001 |
| **Medication for OUD at hospital admission** | 1,508 (17.8%) | 48 (18.1%) | 1,460 (17.8%) | 0.908 |
| **Previously admitted to hospital** | 1,891(22.4%) | 116 (43.8%) | 1,775 (21.7%) | <0.001 |
| **CDPS Score** | 2.5 (1.6) | 3.11 (0.11) | 2.48 (0.02) | <0.001 |

### 3) Model validation

We validated our model using the frameworks suggested by the International Society for Pharmacoeconomics and Outcomes Research and the Society for Medical Decision Making's Good Research Practices Model Validation guidelines (ISPOR-SMDM) [37]. We explored five components of validity: face validity, internal validity, cross validity, predictive validity, and external validity. As suggested, we provide a non-technical description of our model in S3 File.

## Results

There were 8,450 patients admitted at least once with OUD in Oregon from April 2015 through August 2018. A subset of 6,654 patients were seen by January 1st, 2018. Among this subset, at twelve months, 114 (1.7%) participants died from drug-related causes and 408 (6.1%) died from non-drug related causes. Participant demographics of observed data are included in Table 1.

Transition probabilities derived from Bayesian logistic regression models are depicted in Fig 2. In our study, 4% (95% CI = 2%, 6%) of patients admitted at least once for OUD were referred to an ACS in Oregon. Of those, 47% (95% CI = 37%, 57%) engaged in post-discharge OUD care. Of the 96% not seen by an ACS, 20% (95% CI = 16%, 24%) engaged in post-discharge OUD care. The risk of drug-related death at 12 months among patients who engaged in post-discharge OUD care was 3% (95% CI = 0%, 7%) versus 6% (95% CI = 2%, 10%) in patients who did not engage in care. The risk of non-drug related death was 7% (95% CI = 1%, 13%) among patients who engaged in OUD treatment, versus 9% (95% CI = 5%, 13%) for those who did not. For referral to ACS care, the best-fit Bayesian logistic regression model used an 80% confidence interval; for all other models, a sample size of 0.1% fit best (S1 File). All estimates had acceptable Pareto k-diagnostic values. We report posterior intervals for each covariate from Bayesian logistic regression models in S2 File.

### Model validation

**Face validity.** To assess face validity, one researcher (CK) designed the model and received feedback from experts in addiction medicine outside of the study team about the model's face validity. Experts agreed that the model reflected the path of care for patients admitted to hospitals in Oregon with OUD (*structure*). Further, the use of Oregon Medicaid

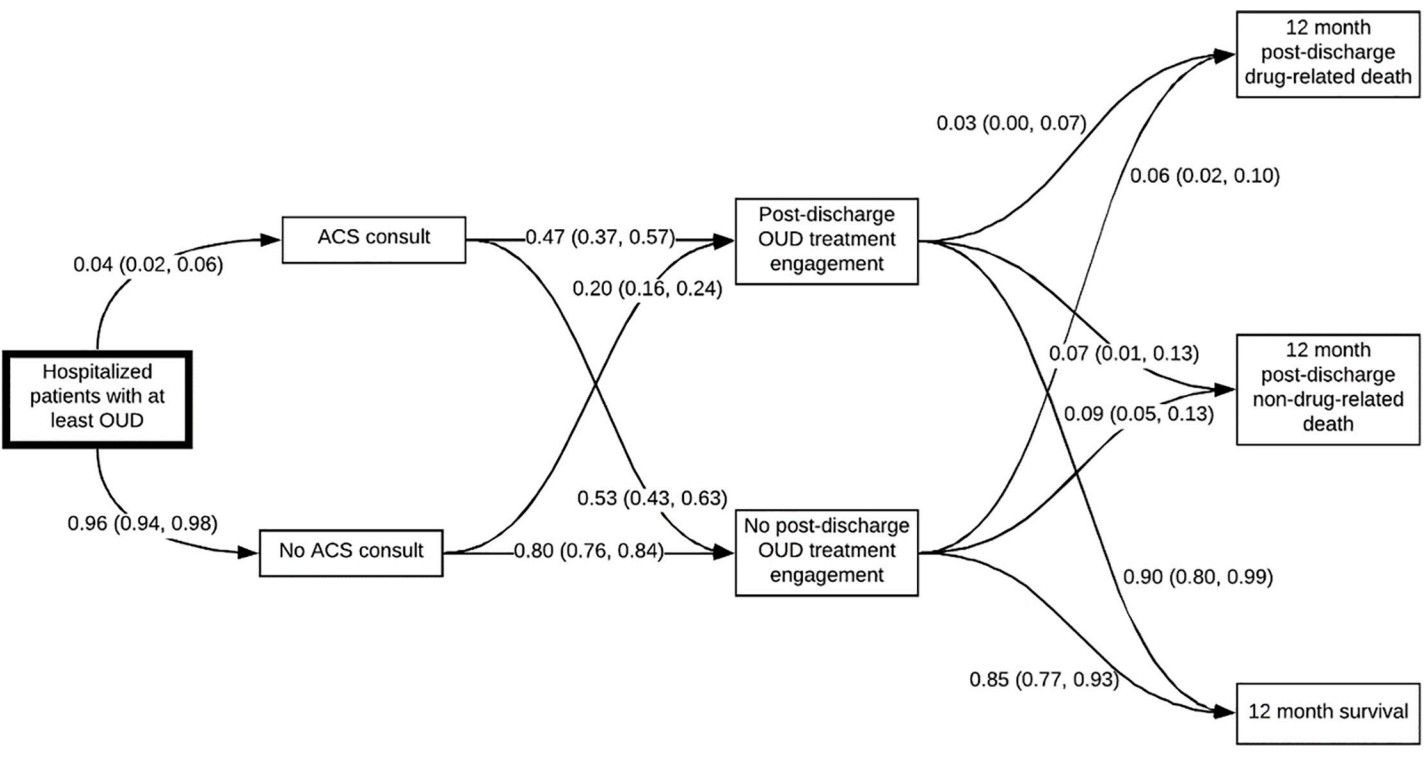

**Fig 2. Markov model with estimated transition probabilities for hospital-based addiction care in Oregon, 2015–2018.**

data, versus data from the literature, was considered a strength in deriving *evidence* for the model by outside experts. ACS and their impact on care for patients with OUD is of immense interest to healthcare systems and policymakers, and experts also agreed that the question was timely and important (*problem formation*). Finally, after data analysis, the model results were presented to researchers who agreed that estimates from the model matched their expectations (*results*).

**Internal validity.** We conducted additional checks and analyses to ensure internal validity of our Bayesian approach (also referred to as technical validity, [38]). First, a recent paper used a similar approach and data structure to evaluate the impact of prenatal maternal factors on nonadherence to infant HIV medication in South Africa. After building our Bayesian model, we used the deidentified data from the South Africa analysis to attempt to replicate identical results as were published. The built model exactly replicated the results of the South African analysis. Second, we conducted classic logistic regression models for each transition point in addition to the Bayesian models. We placed a 1/3, 1/3 noninformative prior (Kerman's prior) on all covariates, which should be roughly approximate to the classic logistic regression results. Our results with non-informative priors were sufficiently similar to classical logistic regression results. Finally, we conducted code "walk throughs" as suggested, where the analyst (CK) walked through code with an expert in these methods (RC).

In addition to the above steps, because we used Bayesian analyses for our transition probabilities, we needed to ensure that our final estimates of confidence intervals around engagement and mortality estimates actually encompassed the observed number of people who engaged, and people who died from drug-related and non-drug related deaths. We simulated estimates, generating "Low" and "High" modeled estimates based on "best" and "worst" cases of model dynamics (e.g. lower confidence bound of estimate for ACS referral, lower

confidence bound for post-discharge OUD treatment engagement, upper confidence bound for drug-related mortality generates an estimate for "High" death). Of the 6,654 patients with 12 months follow-up time, the model estimates that 1,330.8 patients engage in care (Low, High = (1,064.6, 1,597.0)). We observed 1,318 patients who engaged in care in the cohort. Additionally, the model estimated 357.2 drug related deaths (Low, High = (98.5, 632.6)); there were 114 observed drug related deaths in the dataset. Similarly, the model predicted 570.8 non-drug related deaths (Low, High = (263.6, 865.0)); there were 408 observed non-drug related deaths in the dataset. Mortality analyses rarely account for all sources of follow-up which may mean that reported mortality estimates in the literature are lower than in reality. Thus, it was not surprising that modeled transition probabilities from Bayesian logistic regression for 12-month mortality may be higher than raw observed proportions.

**Cross-validation.**   Researchers at a separate academic medical center have developed, validated and calibrated the Reducing Infections Related to Drug Use Cost-Effectiveness (REDUCE) model, a Monte Carlo microsimulation model [39]. This model has the capacity to answer similar questions to what we post here, using estimates derived from published data and from expert sources. In contrast to our model which uses a cohort defined by opioid use disorder, the REDUCE model simulates data for people who inject drugs. Because model estimates for the REDUCE model are derived from a variety of sources in different parts of the county, we expected outcomes from the REDUCE model to be different from our model; we felt these differences are important to understand.

To support cross-validation of our model, the research team that developed the REDUCE model generated 4,153 simulated patients admitted to the hospital for the first time. Of those, 36 died while in the hospital (0.9%). Of the 4117 still alive at hospital discharge, 96 (2.3%) died within 12 months of hospital discharge (95% CI = 1.9%, 2.8%). This is lower than our estimated 928 (13.9%) deaths from our Markov model (Low, High = (5.4%, 22.5%)).

There are several important differences between the REDUCE model and our model. First, as previously mentioned, the REDUCE model simulates data from patients who inject drugs, while ours models patients who have OUD more generally. There are important demographic differences between these two groups, including that our model also includes patients with a primary diagnosis of cancer. Next, the percentage of people seen by an ACS in the REDUCE model was higher than in our model: 25% of patients in REDUCE were seen by an ACS versus 4% in our model. The REDUCE model uses data from Boston, where higher numbers of patients are seen by ACS. This makes it challenging to understand REDUCE estimates in the context of Oregon specifically. Additionally, patients had a higher post-discharge treatment engagement rate in the REDUCE model. In REDUCE, approximately 25.2% of patients receive medication for OUD for at least one week in the month following discharge, versus our model, where 20% of patients not seen by an ACS receive MOUD after discharge. Finally, data from the first simulated admission was used to estimate 12-month mortality from REDUCE; because we matched our cohort controls on the number of previous admissions among patients seen by an ACS, it is possible that our patients were older and sicker than patients who had never previously been admitted to the hospital. While the base model structures are similar, our model is populated with data that provides a focused understanding of addiction consult services in Oregon. Populating our model with different data, including Boston estimates, could provide tailored explorations of ACS in different settings.

**External validity.**   To examine external validity, we used large, high-quality, recent studies of representative populations in independent cohorts of participants to separately validate post-discharge OUD treatment engagement and 12-month drug related and non-drug related mortality. We simulated a cohort of size determined from outside research and looked to see if our simulated confidence interval (cohort simulation/matrix multiplication method, [38]) was

**Table 2. Table of results for external validation of Markov model.**

| Data Source | Justification of selection | Dependent, partially dependent, independent data source | Part of model evaluated | Comparison of differences and results in data sources | Evaluation of cohort simulation results versus observed data |
|---|---|---|---|---|---|
| Naeger et al. [40] | Testing in national dataset | Independent | Post-discharge OUD treatment engagement | Data from 36,719 patients with an inpatient admission for opioid abuse, dependence, or overdose, 2010 to 2014 • Data from time period just prior to Oregon Medicaid cohort; engagement may have been lower • Included any prescription for post-discharge MOUD | Cohort simulation showed 7343.8 (Low, High = (5875, 8812)) people predicted to engage versus 6132 people observed • Modeled range of estimates contains point estimate of observed engagement |
| LaRochelle et al 2018 [41] | Testing in large cohort study | Independent | 12-month drug and non-drug related mortality | 17,568 Massachusetts adults without cancer from 2012 to 2014 • Dataset mortality may be lower because of exclusion of patients with cancer • Post-discharge treatment engagement for OUD included all time, to 12 months, of post-discharge engagement, which may further decrease drug-related deaths | Cohort simulation showed 8.6 non-drug related deaths per 100 person-years (Low, High = (1.5, 13.0)), and 5.4 opioid-related deaths per 100 person-years (Low, High = (4.0, 9.5)) • Observed all-cause mortality was 4.7 deaths (Low, High = (4.4, 5.0)) per 100 person-years; opioid-related mortality was 2.1 deaths (Low, High = (1.9 to 2.4)) per 100-person years • Opioid-related deaths may be higher in our model because of a more liberal definition of opioid-related deaths |
| Ashman et al. (CDC) [42] | Testing in large cohort study | Independent | 12-month all-cause mortality | • 24,340 patients with an opioid hospitalization across 94 National Hospital Care Survey hospitals • Analysis included patients with cancer | Cohort simulation showed 3,394 all-cause deaths (Low, High = (1324, 5478)) versus 1,879 (2,295*0.819) all-cause deaths observed • Modeled range of estimates contains point estimate of observed all-cause mortality |

different from observed values or confidence intervals from the published estimates (Table 2). Where there was disagreement, we describe potential causes.

**Predictive validity.** All relevant data was included in building the Markov model described in this paper. We have planned analyses to evaluate our model predictions versus Medicaid claims data for the same cohort of patients seen in through December of 2020, once data are released.

## Discussion

We built and validated a Markov model that reflects trajectories of care and survival at twelve months for patients hospitalized with OUD in Oregon. We used a Bayesian framework to integrate clinical expertise with data from Oregon Medicaid claims to estimate transition probabilities in our model. After development, we validated our model using ISPOR-SMDM standards, evaluating face validity, internal validity, cross validity, predictive validity and external validity.

Compared to the REDUCE model—another model that assess ACS care delivery—our estimates are more context-relevant estimates of post-discharge OUD treatment engagement and 12-month drug and non-drug related mortality in Oregon. Our overall mortality estimate is higher than the REDUCE model, which may reflect severity of illness of people who are older, sicker, with more previous inpatient hospitalizations and limited linkage to post-discharge OUD care in Oregon. This is important as one potential use of our populated model is to predict the impact of expanding inpatient ACS care in Oregon; a model populated with Oregon data may better reflects the local care setting at baseline may provide more accurate results

following intervention. Additionally, populating our model with different data in different ACS context may similarly provide tailored results.

This study had several limitations. First, because we sought to build a model that reflected addiction care in Oregon, the model may not be generalizable to other settings. Still, the Oregon experience may help inform modeling in other states with limited ACS uptake, and we used Bayesian estimates from national experts to inform transition probabilities. Second, Medicaid claims data are often inaccurate in classifying patient race and ethnicity; our study estimates may not correctly capture the experience of people of color in Oregon. Third, we originally planned to use 30-day mortality as an outcome for this study, but we were unable to do so because of limited drug-related mortality in the 30-day post-discharge period; we used 12-month mortality data instead. Fourth, deriving Bayesian priors from expert elicitation may be less than ideal; clinician estimates may be inaccurate. However, in the absence of published priors available for our transition probabilities, expert elicitation was an appropriate first step to help answer these research questions. Finally, Medicaid claims data does not separate costs for inpatient delivery of medication for OUD, so it was not possible to tell if patients received OUD inpatient outside of an ACS.

This model can be used to evaluate changing scenarios of care in spaces where healthcare providers, healthcare systems, or policymakers are considering implementing or changing ACS coverage in their applicable system. Specifically, this model has been used to evaluate the impact of expanding ACS in Oregon on post-discharge treatment engagement, and to estimate the impact of increasing fentanyl in Oregon's drug supply on post-discharge overdose deaths. Results from these analyses have been submitted to help inform Oregon Medical Association and Oregon Hospital Association decision-making about ACS expansion in the state. The strength of the model comes from the estimates used to populate it, and with recalibration, the model can be adapted to different settings of ACS care delivery. For example, a different hospital with an ACS could estimate transition probabilities by using this model and their own local data. Similarly, hospitals considering ACS expansion could use our Bayesian estimate of ACS effectiveness (derived with a prior from sites across the United States) and combine this with local data for other estimates. This could provide hospitals with an estimate of what might be feasible with ACS implementation. In this paper, we describe data that reflects ACS care in Oregon. Using this data, we can model changing scenarios of care in Oregon, from increasing ACS care delivery to implementing drug-policy related changes, potentially including reducing barriers to naloxone access, implementing safe consumption sites or safe supply interventions, and others. Future research should use this model to evaluate changes in care delivery in Oregon to understand how these changes may impact survival among patients with OUD.

## Conclusion

Hospitalization is a critical time for patients with OUD, and addiction consult services can help support patients during hospitalization and connect them to post-discharge care. Markov modeling can help researchers, clinical teams and policy makers understand how changes in care systems might impact patient outcomes. Additionally, our model allows healthcare systems and policymakers to evaluate the impact of ACS on mortality. In this work, we built and validated a Markov model that reflects the trajectories of care and survival for patients hospitalized with OUD in Oregon. Future research should use this work to evaluate state-wide clinical and policy changes that may impact patient survival.

## Supporting information

**S1 File. Model fit statistics.**
(DOCX)

**S2 File. Estimates from classical and Bayesian logistic regression models, and prior-posterior plots.**
(DOCX)

**S3 File. Non-technical model description.**
(DOCX)

## Acknowledgments

We would like to thank Dr. Christina Nicolaidis, MD, MPH, for her guidance and support for this project.

## Author Contributions

**Conceptualization:** Caroline A. King, Honora Englander, Joshua A. Barocas, K. John McConnell, Cynthia D. Morris, Ryan Cook.

**Data curation:** Caroline A. King.

**Formal analysis:** Caroline A. King, Ryan Cook.

**Funding acquisition:** Caroline A. King.

**Investigation:** Caroline A. King, Ryan Cook.

**Methodology:** Caroline A. King, Honora Englander, P. Todd Korthuis, Joshua A. Barocas, K. John McConnell, Cynthia D. Morris, Ryan Cook.

**Project administration:** Caroline A. King, Honora Englander, P. Todd Korthuis, Cynthia D. Morris, Ryan Cook.

**Resources:** Honora Englander, P. Todd Korthuis, Ryan Cook.

**Supervision:** Honora Englander, P. Todd Korthuis, Cynthia D. Morris, Ryan Cook.

**Validation:** Caroline A. King, Ryan Cook.

**Visualization:** Caroline A. King.

**Writing – original draft:** Caroline A. King.

**Writing – review & editing:** Caroline A. King, Honora Englander, P. Todd Korthuis, Joshua A. Barocas, K. John McConnell, Cynthia D. Morris, Ryan Cook.

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
