## [Decision Letter · Decision Letter 0]

13 Apr 2021

PONE-D-20-37837

Designing and validating a Markov model for hospital-based addiction consult service impact on 12-month drug and non-drug related mortality

PLOS ONE

Dear Dr. King,

Thank you for submitting your manuscript to PLOS ONE, and for your continued patience as we completed the first round of the review process. After careful consideration, we feel that your manuscript has merit but does not fully meet PLOS ONE’s publication criteria as it currently stands. Therefore, we invite you to submit a revised version of the manuscript that addresses the points raised during the review process.

Your manuscript has been assessed by two external experts, who have requested clarification on a number of points regarding the study's methodology and specific contribution to the literature.. 

We look forward to receiving your revised manuscript.

Kind regards,

Dr Joseph Donlan

Senior Editor

PLOS ONE

Journal Requirements:

2. Please provide additional details regarding participant consent.

In the ethics statement in the Methods and online submission information, please ensure that you have specified (a) whether consent was informed and (b) what type you obtained (for instance, written or verbal, and if verbal, how it was documented and witnessed). If your study included minors, state whether you obtained consent from parents or guardians. If the need for consent was waived by the ethics committee, please include this information.

'Dr. Korthuis serves as principal investigator for NIH-funded studies that accept donated study medication from Alkermes (extended-release naltrexone) and Indivior (buprenorphine).'

a. Please confirm that this does not alter your adherence to all PLOS ONE policies on sharing data and materials, by including the following statement: "This does not alter our adherence to  PLOS ONE policies on sharing data and materials.” (as detailed online in our guide for authors http://journals.plos.org/plosone/s/competing-interests).  If there are restrictions on sharing of data and/or materials, please state these.

Please note that we cannot proceed with consideration of your article until this information has been declared.

5. Please ensure that you refer to Figures 3 and 4 in your text as, if accepted, production will need this reference to link the reader to each figure.

Reviewers' comments:

Reviewer's Responses to Questions

**Comments to the Author**

1. Is the manuscript technically sound, and do the data support the conclusions?

Reviewer #1: Partly

Reviewer #2: Partly

2. Has the statistical analysis been performed appropriately and rigorously? 

Reviewer #1: I Don't Know

Reviewer #2: I Don't Know

3. Have the authors made all data underlying the findings in their manuscript fully available?

Reviewer #1: No

Reviewer #2: Yes

4. Is the manuscript presented in an intelligible fashion and written in standard English?

Reviewer #1: Yes

Reviewer #2: Yes

5. Review Comments to the Author

Reviewer #1: The current study aimed to develop a Markov model to estimate the impact of engaging opioid use disorder (OUD) patients in hospital-based addiction consult services (ACS) on drug and non-drug related mortality. The topic is indeed very important given the state of the opioid crisis and the purpose of modeling the effect of such services can be significant in helping other health systems model the impact of creating similar addiction services. My concern is that as an epidemiologist working in the content area but without much experience with such Markov models, I found the explanation of the process and outcomes to be quite confusing. Admittedly, some of my limitations in understanding may result from my own lack of expertise in these statistical models. However, given Plos one has a wide array of readers from different research areas, it is likely others will also require further explanation of this process and I provide some comments below with specific questions/suggestions about the methods and elsewhere.

Intro

The 20% of people who die from a drug overdose is an estimate from one study/region - authors should include the word “an estimated 20%” to clarify this is not necessarily a comprehensive/representative rate.

The second sentence ends quite abruptly with “also contribute.” (contribute to risk of death among this population?)

Given some readers of Plos One may not have a background in addiction treatment, the intro should at least give 1-2 more sentences about what it means to start people on MOUD in the hospital and transition them to ongoing care.

The fourth paragraph should refer to “simulation modeling” rather than modeling so the reader knows what type of modeling is being referred to

Methods

The entire order of the methods section is very confusing. It is not clear what piece of the process came first, what estimates are derived from the Oregon data itself vs. the simulated model or even what data were used for deriving these estimates and the model parameters. I recommend the section begin by describing all the datasets and variables that were used in this analysis and then moving chronologically through the process of generating the model and the estimates. For example, it’s unclear if the authors had data on ACS participation or if this was simulated. It is unclear how these data were linked (if at all) - the hospital ACS information, the Medicaid data, and the mortality data.

The authors call this an “intention to treat” approach - which is a term used for randomized studies. Since this is not a randomized study, nor a quasi experimental study, it seems nonsensical to use this word as the process of whether someone is referred to an ACS does not seem to be random at all.

Related to the prior comment, it is not clear how the models are taking into account the different characteristics of people who do and do not get referred to an ACS - for example, a person referred to an ACS can be at a much more advanced stage of OUD and therefore have higher mortality than someone who was never referred - how is that taken into account? Later the authors mention covariates but there is no explanation of how they get incorporated into the Markov model (again, this may be my limitation in understanding, but an explanation of how/if this comes into play would be helpful)

I am wary of this process by which probability parameters are being estimated from expert opinion - clinicians are generally not good at guessing probabilities from their own experiences so it’s unclear how this is a meaningful source rather than estimates that have been derived in the literature for example? If this is indeed a valid method that has been used in the past to estimate probabilities of this nature, authors should in the least explain this process and cite reference to this method being a valid way to generate these simulations

There needs to be greater explanation of how the expert opinion probabilities were integrated with their actual estimates derived from Oregon data to inform the models ? Again, further explanation of Markov models may generally help readers from different disciplines better understand the methods for this paper and also the added value of using these models here

Results

In the first paragraph, who are the 6,654 patients? those seen by ACS?

Discussion

It would be helpful to include greater discussion around how such modeling adds value - for example, how hospital systems may use such models to make calculations on value of incorporating such programs or how hospitals with ACS could use these modeling techniques to better understand their own population and how they can improve their protocols/systems.

Reviewer #2: PDF file attached. The comments are in the pdf file. The statistical analysis looks complete from the perspective of the data gathered, but it is not clear how this can be used by others in the field, especially by decision-makers in public policy.

6. PLOS authors have the option to publish the peer review history of their article (what does this mean?). If published, this will include your full peer review and any attached files.

Reviewer #1: No

Reviewer #2: No

---

## [Author Response · Author response to Decision Letter 0]

27 May 2021

Reviewer #1: The current study aimed to develop a Markov model to estimate the impact of engaging opioid use disorder (OUD) patients in hospital-based addiction consult services (ACS) on drug and non-drug related mortality. The topic is indeed very important given the state of the opioid crisis and the purpose of modeling the effect of such services can be significant in helping other health systems model the impact of creating similar addiction services. My concern is that as an epidemiologist working in the content area but without much experience with such Markov models, I found the explanation of the process and outcomes to be quite confusing. Admittedly, some of my limitations in understanding may result from my own lack of expertise in these statistical models. However, given Plos one has a wide array of readers from different research areas, it is likely others will also require further explanation of this process and I provide some comments below with specific questions/suggestions about the methods and elsewhere.

Thank you for reviewing our manuscript. We have thoroughly revised the manuscript with regards to your points below, and believe this has improved readability, particularly for audiences from different research areas. 

Intro

1. The 20% of people who die from a drug overdose is an estimate from one study/region - authors should include the word “an estimated 20%” to clarify this is not necessarily a comprehensive/representative rate.

We have made this change (Introduction, paragraph 1). 

2. The second sentence ends quite abruptly with “also contribute.” (contribute to risk of death among this population?)

We have updated to include the following:

“Among people with opioid use disorder (OUD), an estimated 20% eventually die of drug overdose (2), but cardiovascular diseases, cancer, and infectious diseases also contribute to mortality rates.” (Introduction, paragraph 1).

3. Given some readers of Plos One may not have a background in addiction treatment, the intro should at least give 1-2 more sentences about what it means to start people on MOUD in the hospital and transition them to ongoing care.

We have updated to include the following:

”Hospitalization is a vulnerable time for patients with OUD. People with OUD may leave the hospital before completing recommended medical therapy if withdrawal symptoms are untreated (3). People who withdraw from opioids have lower drug tolerance and increased risk of drug overdose after discharge in the absence of treatment for OUD (4-6). Medications for opioid use disorder (MOUD), including methadone, buprenorphine and naltrexone, can reduce the risk of death from opioid overdose in patients with OUD (7). These medications work as opioid receptor full agonists (methadone), partial agonists (buprenorphine), or antagonists (naltrexone) (8). Despite the success of MOUD to reduce opioid overdose deaths, most hospitalized patients with OUD are not started on MOUD (9, 10), though, when offered, nearly three-quarters of patients with OUD choose to start MOUD (11). Interventions to improve initiation of MOUD among hospitalized patients are urgently needed (12).” (Introduction, paragraph 2).

4. The fourth paragraph should refer to “simulation modeling” rather than modeling so the reader knows what type of modeling is being referred to

We have made this change (Introduction, paragraph 4).

Methods

5. The entire order of the methods section is very confusing. It is not clear what piece of the process came first, what estimates are derived from the Oregon data itself vs. the simulated model or even what data were used for deriving these estimates and the model parameters. I recommend the section begin by describing all the datasets and variables that were used in this analysis and then moving chronologically through the process of generating the model and the estimates. For example, it’s unclear if the authors had data on ACS participation or if this was simulated. It is unclear how these data were linked (if at all) - the hospital ACS information, the Medicaid data, and the mortality data. Related to the prior comment, it is not clear how the models are taking into account the different characteristics of people who do and do not get referred to an ACS - for example, a person referred to an ACS can be at a much more advanced stage of OUD and therefore have higher mortality than someone who was never referred - how is that taken into account? Later the authors mention covariates but there is no explanation of how they get incorporated into the Markov model (again, this may be my limitation in understanding, but an explanation of how/if this comes into play would be helpful)

Thank you for this very helpful comment. We have reorganized the methods to make our order of steps clear. For example, we now state:

“We developed and validated a Markov model to estimate the impact of ACS care on 12-month mortality among hospitalized patients with OUD (Fig 1). We organize our methods in the order of completion: first, we decided on model structure, next we used available data to populate the model, and last, we validated the model. As such, we describe: 1) model structure, 2) model data, and 3) model validation.” (Methods Paragraph 2). 

We then number to delinate each step, and then provide additional introduction sections where we thought there was confusion. For example, at the beginning of the model data step, we wrote the following, which also highlights the integration of observed ACS data, Oregon Medicaid data and Vital Statistics data. 

“We had multiple goals in using data to populate our model. We needed a dataset of patients, where some patients were referred to ACS and some were not. Then, we needed to be able to match ACS patients to controls as one way to account for some confounding. We needed the dataset to follow both patients referred to ACS, and those not, through 12 months after hospital discharge. Finally, we needed the dataset to have additional covariates to control for additional confounding, which we planned to do via logistic regression models at each transition point. Below, we describe the integration of OHSU’s ACS dataset with Oregon Medicaid data and Vital Statistics data, to achieve our goal for a dataset for model population. Because we wanted to incorporate national estimates into our dataset, we used Bayesian logistic regression to integrate expert opinion into our estimates. We describe each of these steps below.” (Methods Paragraph 8)

We have also clarified where data is from simulated vs observed estimates, for example,

“Among this subset, at twelve months, 114 (1.7%) participants died from drug-related causes and 408 (6.1%) died from non-drug related causes. Participant demographics of observed data are included in Table 1.” (Results Paragraph 1)

We included the following sentence to highlight that ACS referral data was observed:

“Because OHSU IMPACT was the only ACS in Oregon during the study window, we used IMPACT registry which tracks all referrals to identify patients with Oregon Medicaid who were referred to ACS.” (Methods Paragraph 10)

Confounding was accounted for by extracting estimates from logistic regression models that accounted for confounders. So, for example, we used logistic regression to evaluate the impact of post-discharge MOUD on 12-month mortality, accounting for confounders. From the adjusted model, we extracted the marginal probability of post-discharge MOUD; this is our base transition probability. The Bayesian model uses the national estimate from experts as our prior, and integrates the marginal probability from the logistic regression model, to give us an updated posterior probability that incorporates both estimates. We have updated the manuscript to better describe this for a lay audience:

“We used a Bayesian approach to obtain transition probabilities for our Markov model using Oregon data an describe this below. In short, we wanted to integrate expert information nationally with estimates from Oregon data for each of our three transition steps: ACS referral, post-discharge MOUD, and 12-month mortality. We also needed to adjust for confounding at each transition point. Bayesian logistic regression allowed us to accomplish this goal. We were able to run logistic regression models for each transition point, using the transition as the outcome (i.e. an outcome of 12-month post-discharge mortality) and the prior step as the primary covariate of interest (i.e. 30-day post-discharge MOUD), adjusting for all other covariates in the model. We extract a marginal probability from this logistic regression model- this is our Bayesian likelihood. We used information from experts in addiction (described below) as our prior. The Bayesian approach allows the integration of the prior and likelihood to estimate a posterior probability, which we use as our posterior probability.”(Methods, Paragraph 12)

Length of time with OUD was not available as a confounder, and is not possible to accurately calculate from Oregon Medicaid data. 

6. The authors call this an “intention to treat” approach - which is a term used for randomized studies. Since this is not a randomized study, nor a quasi experimental study, it seems nonsensical to use this word as the process of whether someone is referred to an ACS does not seem to be random at all.

Thank you- we agree. We have updated to read as follows:

“For this model, all patients referred to ACS were included regardless of level of care engagement or specific services received.” (Methods, Paragraph 2). 

7. I am wary of this process by which probability parameters are being estimated from expert opinion - clinicians are generally not good at guessing probabilities from their own experiences so it’s unclear how this is a meaningful source rather than estimates that have been derived in the literature for example? If this is indeed a valid method that has been used in the past to estimate probabilities of this nature, authors should in the least explain this process and cite reference to this method being a valid way to generate these simulations

Thank you for this comment. ACS are a novel intervention, and prior to this manuscript, there were no published estimates of the parameters we sought to evaluate for ACS involvement. Thus, we chose to use expert elicitation to derive study estimates. Often models rely on expert opinion when there is a lack of available data to inform model parameters. We agree that expert opinion may be less than optimal, however, the Bayesian process we used helps account for some of the uncertainty that comes from incorporating expert opinion. We have updated the methods and limitations to describe this (Methods, Paragraph 11; Discussion, Paragraph 3). 

“Because of the novelty of ACS, few published papers existed from which we could have derived prior estimates of transition probabilities for Bayesian analysis. Thus, we used expert elicitation to capture prior information for our models. The Bayesian process helped account for some for some of the uncertainty.”(Methods, Paragraph 11)

“Fourth, deriving Bayesian priors from expert elicitation may be less than ideal; clinician estimates may be inaccurate. However, in the absence of published priors available for our transition probabilities, expert elicitation was an appropriate first step to help answer these research questions.” (Discussion, Paragraph 3)

8. There needs to be greater explanation of how the expert opinion probabilities were integrated with their actual estimates derived from Oregon data to inform the models ? Again, further explanation of Markov models may generally help readers from different disciplines better understand the methods for this paper and also the added value of using these models here

Thank you. We have updated to include the following:

“We used a Bayesian approach to obtain transition probabilities for our Markov model using Oregon data an describe this below. In short, we wanted to integrate expert information nationally with estimates from Oregon data for each of our three transition steps: ACS referral, post-discharge MOUD, and 12-month mortality. We also needed to adjust for confounding at each transition point. Bayesian logistic regression allowed us to accomplish this goal. We were able to run logistic regression models for each transition point, using the transition as the outcome (i.e. an outcome of 12-month post-discharge mortality) and the prior step as the primary covariate of interest (i.e. 30-day post-discharge MOUD), adjusting for all other covariates in the model. We extract a marginal probability from this logistic regression model- this is our Bayesian likelihood. We used information from experts in addiction (described below) as our prior. The Bayesian approach allows the integration of the prior and likelihood to estimate a posterior probability, which we use as our posterior probability.”(Methods, Paragraph 12)

Results

9. In the first paragraph, who are the 6,654 patients? those seen by ACS?

Thank you- this sentence was confusing. We have rewritten as follows:

“There were 8,450 patients admitted at least once with OUD in Oregon from April 2015 through August 2018. A subset of 6,654 patients were seen by January 1st, 2018. Among this subset, at twelve months, 114 (1.7%) participants died from drug-related causes and 408 (6.1%) died from non-drug related causes.” (Results, Paragraph 1)

Discussion

10. It would be helpful to include greater discussion around how such modeling adds value - for example, how hospital systems may use such models to make calculations on value of incorporating such programs or how hospitals with ACS could use these modeling techniques to better understand their own population and how they can improve their protocols/systems.

We have updated the final paragraph of the Discussion to read as follows:

“This model can be used to evaluate changing scenarios of care in spaces where healthcare providers, healthcare systems, or policymakers are considering implementing or changing ACS coverage in their applicable system. Specifically, this model has been used to evaluate the impact of expanding ACS in Oregon on post-discharge treatment engagement, and to estimate the impact of increasing fentanyl contamination in Oregon’s drug supply on post-discharge overdose deaths. Results from these analyses have been submitted to help inform Oregon Medical Association and Oregon Hospital Association decision-making about ACS expansion in the state. The strength of the model comes from the estimates used to populate it, and with recalibration, the model can be adapted to different settings of ACS care delivery. For example, a different hospital with an ACS could estimate transition probabilities by using this model and their own local data. Similarly, hospitals considering ACS expansion could use our Bayesian estimate of ACS effectiveness (derived with a prior from sites across the United States) and combine this with local data for other estimates. This could provide hospitals with an estimate of what might be feasible with ACS implementation. In this paper, we describe data that reflects ACS care in Oregon. Using this data, we can model changing scenarios of care in Oregon, from increasing ACS care delivery to implementing drug-policy related changes, potentially including reducing barriers to naloxone access, implementing safe consumption sites or safe supply interventions, and others. Future research should use this model to evaluate changes in care delivery in Oregon to understand how these changes may impact survival among patients with OUD.” (Discussion, Paragraph 4). 

11. Reviewer #2: The statistical analysis looks complete from the perspective of the data gathered, but it is not clear how this can be used by others in the field, especially by decision-makers in public policy.

Thank you. We have updated the Discussion; please see comment 10 above from Reviewer #1. Additionally, we have included Appendix 3, which is a lay-summary for healthcare policy makers who may wish to use the model to estimate policy changes.

12. The paper presents a Markov-chain based model to analyze data from patients suffering from OUD. A Bayesian regression-based model is used to estimate the transition probabilities of different stages. The data set is quite large, and the work is described well. While I do believe this work will benefit decision makers, the authors definitely need to do some additional analysis of what the impact is of the consultation service. What is the difference in mortality when they take the consultation service and/ or do not use the post-discharge treatment? Without this analysis, it is unclear what gap in the literature is filled and what the motivation is for this research: it just provides some data, but not clear how it may be used by decision-makers in public policy. It is possible to use steady-state probabilities to determine the long-run survival and mortality rates (see e.g., Gosavi et al., 2020). Further, even some basic analysis is possible from the probabilities in Fig 2: what is the probability that someone who took advantage of the consulting service survived, vs that of someone who did not? 

Thank you. We agree. We had two goals for this work: to derive a simulation model using Bayesian techniques that could estimate post-discharge mortality, and to derive estimates of mortality under different conditions to answer pertinent health policy questions. This manuscripts addresses the first question. To answer the second, we conducted several other analyses, which we have described in publications that have been submitted elsewhere. These manuscripts estimate changes in treatment engagement and mortality outcomes in the context of ACS engagement. The first explores the impact of ACS expansion through Medicaid Coordinated Care Organizations in Oregon, on post-discharge treatment engagement. The second evaluates the impact of increasing fentanyl contamination in Oregon on post-discharge mortality, and elucidates the role ACS expansion could play in decreasing mortality. We originally considered combining all estimates into one paper. However, the work in this manuscript is itself novel in constructing the model, and, for maximum impact, we wanted to write the additional manuscripts with only as much technical information as was necessary to interpret model results. To achieve this balance, we decided to submit this manuscript as a model development and validation paper, with subsequent manuscripts geared at using the model to answer pertinent policy questions. In case of interest, abstracts from these two additional manuscripts are as follows: 

Expanding inpatient Addiction Consult Services through Accountable Care Organizations for Medicaid enrollees: A modeling study 

Introduction

Addiction Consult Services (ACS) care for patients with opioid use disorder (OUD) in the hospital. Medicaid Accountable Care Organizations (ACOs) could enhance access to ACS. This study extends data from Oregon's only ACS to Oregon's 15 regional Medicaid Coordinated Care Organizations (CCOs) to illustrate the potential value of enhanced in- and out-patient care for hospitalized patients with OUD. The study objectives were to estimate the effects of 1) expanding ACS care through CCOs in Oregon, and 2) increasing community treatment access within CCOs, on post-discharge OUD treatment engagement. 

Methods

We used a validated Markov model, populated with Oregon Medicaid data from April 2015 to December 2017, to estimate study objectives.

Results

Oregon Medicaid patients hospitalized with OUD with care billed to a CCO (n=5,878) included 1,298 (22.1%) patients engaged in post-discharge OUD treatment. Simulation of referral to an ACS increased post-discharge OUD treatment engagement to 47.0% (95% CI 45.7%, 48.3%), or 2,684 patients (95% CI 2610, 2758). Ten of fifteen (66.7%) CCOs had fewer than 20% of patients engage in post-discharge OUD care. Without ACS, increasing outpatient treatment such that 20% of patients engage increased the patients engaging in post-discharge OUD care from 12.9% or 296 patients in care at baseline to 20% (95% CI 18.1%, 21.4%) or 453 (95% CI 416, 491).

Discussion

ACOs can improve care and coordination for patients hospitalized with OUD. Implementing ACS in ACO networks can potentially improve post-discharge OUD treatment engagement, but community treatment systems must be prepared to accept more patients as inpatient addiction care improves. 

Hospitalizations, drug supply contamination, and 12-month post-discharge drug-related mortality among patients with opioid use disorder: modeling the role of Addiction Consult Services 

Introduction

Addiction Consult Services (ACS) care for patients with opioid use disorder (OUD) in the hospital. Medicaid Accountable Care Organizations (ACOs) could enhance access to ACS. This study extends data from Oregon's only ACS to Oregon's 15 regional Medicaid Coordinated Care Organizations (CCOs) to illustrate the potential value of enhanced in- and out-patient care for hospitalized patients with OUD. The study objectives were to estimate the effects of 1) expanding ACS care through CCOs in Oregon, and 2) increasing community treatment access within CCOs, on post-discharge OUD treatment engagement. 

Methods

We used a validated Markov model, populated with Oregon Medicaid data from April 2015 to December 2017, to estimate study objectives.

Results

Oregon Medicaid patients hospitalized with OUD with care billed to a CCO (n=5,878) included 1,298 (22.1%) patients engaged in post-discharge OUD treatment. Simulation of referral to an ACS increased post-discharge OUD treatment engagement to 47.0% (95% CI 45.7%, 48.3%), or 2,684 patients (95% CI 2610, 2758). Ten of fifteen (66.7%) CCOs had fewer than 20% of patients engage in post-discharge OUD care. Without ACS, increasing outpatient treatment such that 20% of patients engage increased the patients engaging in post-discharge OUD care from 12.9% or 296 patients in care at baseline to 20% (95% CI 18.1%, 21.4%) or 453 (95% CI 416, 491).

Discussion

ACOs can improve care and coordination for patients hospitalized with OUD. Implementing ACS in ACO networks can potentially improve post-discharge OUD treatment engagement, but community treatment systems must be prepared to accept more patients as inpatient addiction care improves. 

13. Typo: In Discussion: “Second, claims data is often inaccurate in classifying patient race and ethnicity” → Do you mean “…claims that data…? There is a grammar error here. Please fix. Also, please note that data are and not is. 

Thank you. We were referring to Medicaid claims data and have clarified. We have also changed from data “is” to data “are” throughout. (Discussion, Paragraph 3). 

14. Gosavi, A, Murray, S.L., and Karagiannis, N. (2020). A Markov Chain Approach for Forecasting Progression of Opioid Addiction. Proceedings of the Industrial and Systems Engineering Annual Conference, Virtual, L. Cromarty, R. Shirwaiker, P. Wang, eds.

Thank you for this helpful citation- we have added this where appropriate (for example, Introduction, paragraph 4).

---

## [Decision Letter · Decision Letter 1]

17 Aug 2021

Designing and validating a Markov model for hospital-based addiction consult service impact on 12-month drug and non-drug related mortality

PONE-D-20-37837R1

Dear Dr. King,

We’re pleased to inform you that your manuscript has been judged scientifically suitable for publication and will be formally accepted for publication once it meets all outstanding technical requirements.

Kind regards,

Sungwoo Lim, DrPH

Academic Editor

PLOS ONE

Additional Editor Comments (optional):

Please make sure to include a summary about validation findings in the abstracts as well as discussion section.  

Reviewers' comments:

Reviewer's Responses to Questions

**Comments to the Author**

1. If the authors have adequately addressed your comments raised in a previous round of review and you feel that this manuscript is now acceptable for publication, you may indicate that here to bypass the “Comments to the Author” section, enter your conflict of interest statement in the “Confidential to Editor” section, and submit your "Accept" recommendation.

Reviewer #1: All comments have been addressed

2. Is the manuscript technically sound, and do the data support the conclusions?

Reviewer #1: Yes

3. Has the statistical analysis been performed appropriately and rigorously? 

Reviewer #1: Yes

4. Have the authors made all data underlying the findings in their manuscript fully available?

Reviewer #1: Yes

5. Is the manuscript presented in an intelligible fashion and written in standard English?

Reviewer #1: Yes

6. Review Comments to the Author

Reviewer #1: The authors have addressed my concerns and the manuscript is significantly improved. My only suggestion is that authors refer to their other policy-oriented papers that came out of this work in the manuscript so readers can know where to find them.

7. PLOS authors have the option to publish the peer review history of their article (what does this mean?). If published, this will include your full peer review and any attached files.

Reviewer #1: No

---

## [Editor Report · Acceptance letter]

31 Aug 2021

PONE-D-20-37837R1 

Designing and validating a Markov model for hospital-based addiction consult service impact on 12-month drug and non-drug related mortality 

Dear Dr. King:

I'm pleased to inform you that your manuscript has been deemed suitable for publication in PLOS ONE. Congratulations! Your manuscript is now with our production department. 

Kind regards, 

on behalf of

Dr. Sungwoo Lim 

Academic Editor

PLOS ONE